# Pragmatic Image Compression
# for Human-in-the-Loop Decision-Making

**Siddharth Reddy, Anca D. Dragan, Sergey Levine**
University of California, Berkeley
{sgr,anca,svlevine}@berkeley.edu

## Abstract

Standard lossy image compression algorithms aim to preserve an image's appearance, while minimizing the number of bits needed to transmit it. However, the amount of information actually needed by a user for downstream tasks – e.g., deciding which product to click on in a shopping website – is likely much lower. To achieve this lower bitrate, we would ideally only transmit the visual features that drive user behavior, while discarding details irrelevant to the user's decisions. We approach this problem by training a compression model through human-in-the-loop learning as the user performs tasks with the compressed images. The key insight is to train the model to produce a compressed image that induces the user to take the same action that they would have taken had they seen the original image. To approximate the loss function for this model, we train a discriminator that tries to distinguish whether a user's action was taken in response to the compressed image or the original. We evaluate our method through experiments with human participants on four tasks: reading handwritten digits, verifying photos of faces, browsing an online shopping catalogue, and playing a car racing video game. The results show that our method learns to match the user's actions with and without compression at lower bitrates than baseline methods, and adapts the compression model to the user's behavior: it preserves the digit number and randomizes handwriting style in the digit reading task, preserves hats and eyeglasses while randomizing faces in the photo verification task, preserves the perceived price of an item while randomizing its color and background in the online shopping task, and preserves upcoming bends in the road in the car racing game.

## 1 Introduction

Modern web platforms serve billions of images every day, and typically rely on lossy compression algorithms to store and transmit this data efficiently. Recent work on machine learning methods for lossy image compression [1, 2, 3, 4, 5, 6, 7, 8, 9, 10, 11] improves upon standard methods like JPEG [12] by training neural networks to minimize the number of bits needed to generate high-fidelity reconstructions. In this paper, we explore the idea of compressing images to even smaller sizes by intentionally allowing reconstructions to deviate drastically from the visual appearance of their originals, and instead optimizing reconstructions for the specific, downstream tasks that the user wants to perform with them, such as video conferencing, online gaming, or remotely operating space robots [13].

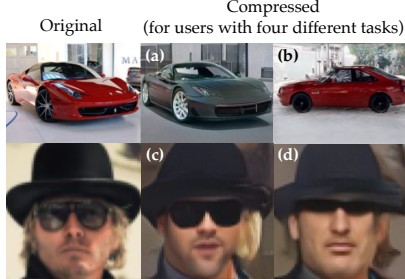

Figure 1: Images compressed 2-4x smaller than JPEG retain information for tasks like shopping for cars in a perceived price range (a), surveying car colors (b), and checking photos for eyeglasses (c) or hats (d).

35th Conference on Neural Information Processing Systems (NeurIPS 2021).

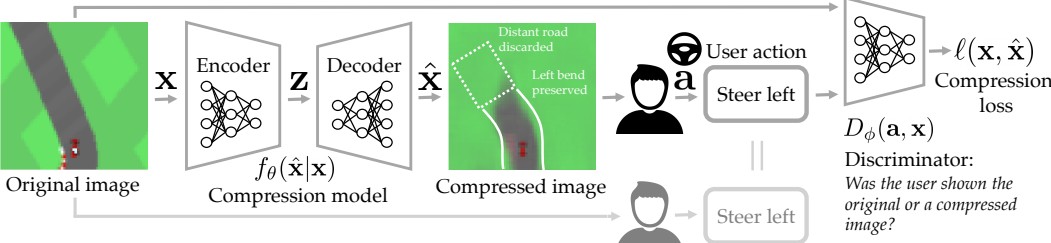

Figure 2: Given the original image $\mathbf{x}$, we would like to generate a compressed image $\hat{\mathbf{x}}$ such that the user's action $\mathbf{a}$ upon seeing the compressed image is similar to what it would have been had the user seen the original image instead. In a 2D top-down car racing video game, our compression model learns that, in order to match the user's steering with and without compression, it must preserve bends, but can discard the road farther ahead.

Our main observation in this work is that, instead of optimizing the compression model for a task-agnostic *perceptual* similarity objective function, we can instead optimize the compression model for *functional* similarity: producing compressed images that, when shown to the user, induce the user to take the same actions that they would have taken had they observed the original, uncompressed images. We call this *PragmatIc COmpression* (PICO), inspired by prior work on pragmatics [14, 15, 16] that characterizes the meaning of a message through the behavior it induces in a listener. PICO adapts compression to user behavior, enabling the user to perform their individually-desired tasks with compressed images instead of the original images. For example, consider two users with distinct tasks: one flying a quadcopter, and the other driving a ground robot. On a network with an extremely low bitrate, we would like the compressed video feed of the ground robot to preserve ground-level obstacles and terrain while discarding details about power lines and tree canopies, and the quadcopter feed to do the opposite.

To this end, we formulate compression as a human-in-the-loop learning problem, in which the compression model is represented as an encoder-decoder neural network that takes the original image as an input and outputs the compressed image. The user sees the compressed image, and takes an action to perform their desired task (see Figure 2). The main challenge in this work is designing a loss function for the compression model that evaluates the quality of the compressed image in the context of the original image and the user's action. We do not assume knowledge of the user's desired task, so we cannot directly evaluate the quality of the compressed image by evaluating the fitness of the user's action upon seeing the compressed image. We also do not assume access to ground-truth action labels for the original images in the streaming setting, so we cannot compare the user's action upon seeing the compressed image to some ground-truth action.

Instead, we define the loss function through adversarial learning. For example, consider a user browsing an online shopping catalogue, observing photos and clicking on appealing items. To collect positive and negative examples of user behavior, we simply randomize whether a user sees the original or compressed version of an image while they are shopping, and record their actions. We then train a discriminator to predict the likelihood that a user's action was taken in response to the original rather than a compressed image, and train the compression model to maximize this predicted likelihood.

Our primary contribution is the PICO algorithm for human-in-the-loop learning of data compression models. We validate PICO through three user studies on Amazon Mechanical Turk, in which we train and evaluate our compression models on data from human participants. In the first study, we asked participants to read handwritten digits and identify the numbers – PICO learned to preserve the number and discard handwriting style (Figure 3). In the second study, we asked users to browse a car catalogue and select cars based on perceived price – PICO learned to preserve overall shape and sportiness while randomizing paint jobs and backgrounds (Figure 4). In the third study, we asked participants to verify photos of faces by checking if heads or eyes were covered – PICO learned to preserve hats and eyeglasses while randomizing faces (Figure 5). In all three studies, PICO obtained up to 2-4x lower bitrates than non-adaptive baseline methods. To show that PICO can be used in sequential decision-making problems, we also ran a user study with 12 participants who played a car racing video game – at a fixed bitrate, PICO learned to preserve bends in the road substantially better than a non-adaptive baseline method, enabling users to drive more safely (Figure 6).

## 2   Related Work

Prior work on learned lossy image compression focuses on overcoming various challenges in training neural networks on images [17], including amortized variable-rate compression [1, 4], end-to-end training with quantization [2, 5, 6], optimizing the rate-distortion trade-off [7, 8], optimizing perceptual quality [9, 10, 11], training hierarchical latent variable models [3], and sequential compression of videos [18, 19]. While these methods aim to generate visually-pleasing reconstructions that are perceptually similar to their originals, PICO focuses on preserving functional similarity. Hence, PICO can achieve substantially lower bitrates for specific downstream tasks (e.g., see Figure 3).

Prior work has studied human-in-the-loop learning in related contexts, including reinforcement learning of text summarization policies from user feedback [20] and automatic data visualization for decision support systems [21]. In the context of imitation learning, the idea of fitting a model of human behavior using generative adversarial networks [22] has also been explored [23]. PICO differs from [21, 23] in that it tackles image compression – an entirely different problem from decision support and imitation learning. In contrast to [20], which elicits user comparisons between different summaries of the same text, PICO can be used for sequential tasks like video games (see Section 5.3) where the user cannot be repeatedly queried with different compressed versions of the same image.

## 3   Pragmatic Compression

Generative models are typically used for sampling and representation learning, but they can also be used for compression [24, 25, 26, 27]. For example, variational autoencoders [28] are trained with a variational information bottleneck [29] that explicitly constrains the amount of information carried by their latent variables – hence, we can use a trained encoder to compress an image, and a trained decoder to reconstruct it from latent features [5, 30]. In contrast to compression methods that train such generative models to maximize the visual fidelity of the reconstruction, we formulate compression as a problem of *control*, including the downstream behavior of the user in the problem statement. First, the environment generates an image $\mathbf{x} \in \mathbb{R}^{w \times h \times c}$. Given the original image $\mathbf{x}$, the compression system generates a compressed image $\hat{\mathbf{x}} \in \mathbb{R}^{w \times h \times c}$ that can be represented using no more than $n$ bits, where $n$ is a hyperparameter. The user then observes the compressed image $\hat{\mathbf{x}}$ and samples an action $\mathbf{a} \sim \pi(\mathbf{a}|\hat{\mathbf{x}})$ from their unknown policy $\pi$. We do not assume access to the user's utility function $U(\mathbf{x}, \mathbf{a})$ or a specification of their desired task. Our goal is to generate a compressed image $\hat{\mathbf{x}}$ that induces an action $\mathbf{a}$ that maximizes the unknown utility $U(\mathbf{x}, \mathbf{a})$.

We approach this problem by generating a compressed image $\hat{\mathbf{x}}$ that induces the user to take the same action $\mathbf{a}$ that they would have taken had they seen the original image $\mathbf{x}$ instead. Let $f_\theta(\hat{\mathbf{x}}|\mathbf{x})$ denote a parametric model of our compression function, where $\theta$ are the model parameters (e.g., neural network weights). To train $f_\theta$, we need a loss function that evaluates the difference between an original image $\mathbf{x}$ and the output of the compression model $\hat{\mathbf{x}} \sim f_\theta(\hat{\mathbf{x}}|\mathbf{x})$. One approach is to use conditional generative adversarial networks [31] to train a discriminator $D(\hat{\mathbf{x}}, \mathbf{x})$ that tries to distinguish between original and compressed images, and train the compression model to generate compressed images $\hat{\mathbf{x}}$ that fool this discriminator, analogous to prior work on adversarial image compression [11]. However, this approach seeks to maximize the *perceptual* similarity of the original and compressed image, whereas we would like to maximize their *functional* similarity.

The key challenge for our method then is to train the discriminator $D(\hat{\mathbf{x}}, \mathbf{x})$ to detect differences between $\mathbf{x}$ and $\hat{\mathbf{x}}$ that influence the user's downstream action, while ignoring superficial differences between the images that do not affect the user's action. We address this challenge by first training an action discriminator $D_\phi(\mathbf{a}, \mathbf{x})$ to predict whether the user saw the original or a compressed image before taking the action $\mathbf{a}$. This action discriminator $D_\phi$ captures differences in user behavior caused by compression, while ignoring visual differences between the original and compressed images. To construct a loss function that links the compressed images to these behavioral differences, we distill the action discriminator $D_\phi(\mathbf{a}, \mathbf{x})$ into an image discriminator $D_\psi(\hat{\mathbf{x}}, \mathbf{x})$.

### 3.1   Maximizing Functional Similarity of Images through Adversarial Learning

We formalize the idea of maximizing the functional similarity of the original $\mathbf{x}$ and compressed image $\hat{\mathbf{x}}$ as follows. Let $T \in \{0, 1\}$ denote whether the user sees the original or a compressed image before taking an action: if $T = 1$, then $\hat{\mathbf{x}} \leftarrow \mathbf{x}$; else if $T = 0$, sample $\hat{\mathbf{x}} \sim f_\theta(\hat{\mathbf{x}}|\mathbf{x})$. We would like to train

---

**Algorithm 1** Pragmatic Compression (PICO)

---

Initialize compression model $f_\theta$
**while** true **do**
    $\mathbf{x} \sim p_{\text{env}}(\mathbf{x})$                                  ▷ environment generates original image
    $T \sim \text{Bernoulli}(0.5)$              ▷ randomly decide whether user sees compressed image or original
    **if** $T = 1$ **then** $\hat{\mathbf{x}} \leftarrow \mathbf{x}$ **else** $\hat{\mathbf{x}} \sim f_\theta(\hat{\mathbf{x}}|\mathbf{x})$
    $\mathbf{a} \sim \pi(\mathbf{a}|\hat{\mathbf{x}})$                                    ▷ user takes action using unknown policy
    $\mathcal{D} \leftarrow \mathcal{D} \cup \{(T, \mathbf{x}, \hat{\mathbf{x}}, \mathbf{a})\}$
    $\phi \leftarrow \phi + \nabla_\phi \sum_{(T,\mathbf{x},\mathbf{a})\in\mathcal{D}} T \cdot \log D_\phi(\mathbf{a}, \mathbf{x}) + (1 - T) \cdot \log (1 - D_\phi(\mathbf{a}, \mathbf{x}))$ ▷ update action discrim.
    $\psi \leftarrow \psi - \nabla_\psi \sum_{(\mathbf{x},\hat{\mathbf{x}},\mathbf{a})\in\mathcal{D}} D_{\text{KL}}(D_\phi(\mathbf{a}, \mathbf{x}) \parallel D_\psi(\hat{\mathbf{x}}, \mathbf{x}))$          ▷ update image discriminator
    $\theta \leftarrow \theta + \nabla_\theta \sum_{\mathbf{x}\in\mathcal{D}} \log D_\psi(f_\theta(\mathbf{x}), \mathbf{x})$                    ▷ update compression model

---

the compression model to minimize the divergence of the user's policy evaluated on the compressed image $\pi(\mathbf{a}|\hat{\mathbf{x}})$ from the policy evaluated on the original $\pi(\mathbf{a}|\mathbf{x})$,

$$\begin{aligned}\mathcal{L}(\theta) &= \mathbb{E}_\mathbf{x}[D(\pi(\mathbf{a}|\mathbf{x}) \parallel \mathbb{E}_{\hat{\mathbf{x}}\sim f_\theta(\hat{\mathbf{x}}|\mathbf{x})}[\pi(\mathbf{a}|\hat{\mathbf{x}})|\mathbf{x}])] \\ &= \mathbb{E}_\mathbf{x}[D(p(\mathbf{a}|\mathbf{x}, T = 1) \parallel p(\mathbf{a}|\mathbf{x}, T = 0; \theta))],\end{aligned} \tag{1}$$

where $D$ is a divergence (e.g., the Jensen-Shannon divergence) – note that we are overloading $D$ to denote a divergence in Equation 1, and to denote a discriminator elsewhere. Since the user's policy $\pi$ is unknown, we approximately minimize the loss in Equation 1 using conditional generative adversarial networks (GAN) [31], where the side information is the original image $\mathbf{x}$, the generator is the compression model $f_\theta(\hat{\mathbf{x}}|\mathbf{x})$, and the discriminator $D(\mathbf{a}, \mathbf{x})$ tries to discriminate the action $\mathbf{a}$ that the user takes after seeing the generated image $\hat{\mathbf{x}}$.

To train the action discriminator, we need positive and negative examples of user behavior; in our case, examples of user behavior with and without compression. To collect these examples, we randomize whether the user sees the compressed image or the original before taking an action. Let $T \sim \text{Bernoulli}(0.5)$ represent this random assignment. When $T = 1$, the user sees the original $\mathbf{x}$ and takes action $\mathbf{a}$, and we record $(\mathbf{x}, \hat{\mathbf{x}}, \mathbf{a})$ as a positive example of user behavior. When $T = 0$, the user sees the compressed image $\hat{\mathbf{x}}$ and takes action $\mathbf{a}$, and we record $(\mathbf{x}, \hat{\mathbf{x}}, \mathbf{a})$ as a negative example. Let $\mathcal{D}$ denote the dataset of all recorded tuples $(T, \mathbf{x}, \hat{\mathbf{x}}, \mathbf{a})$. We train an action discriminator $D_\phi(\mathbf{a}, \mathbf{x})$ to predict the likelihood $p(T = 1|\mathbf{a}, \mathbf{x})$, using the standard binary cross-entropy loss and the training data $\mathcal{D}$. Note that this action discriminator is conditioned on the original image $\mathbf{x}$ and the user action $\mathbf{a}$, but not the compressed image $\hat{\mathbf{x}}$ – this follows from our problem formulation in Equation 1, and ensures that the action discriminator captures differences in user behavior caused by compression, while ignoring differences between the original and compressed images that do not affect user behavior.

### 3.2 Distilling the Discriminator and Training the Compression Model

The action discriminator $D_\phi(\mathbf{a}, \mathbf{x})$ gives us a way to approximately evaluate the loss function in Equation 1. However, we cannot train the compression model $f_\theta(\hat{\mathbf{x}}|\mathbf{x})$ to optimize this loss directly, since $D_\phi$ does not take the compressed image $\hat{\mathbf{x}}$ as input. To address this issue, we distill the trained action discriminator $D_\phi(\mathbf{a}, \mathbf{x})$, which captures differences in user behavior caused by compression, into an image discriminator $D_\psi(\hat{\mathbf{x}}, \mathbf{x})$ that links the compressed images to these behavioral differences. In particular, we train $D_\psi$ to approximate $D_\phi$ by optimizing the loss,

$$\ell(\psi) = \sum_{(\mathbf{x},\hat{\mathbf{x}},\mathbf{a})\in\mathcal{D}} D_{\text{KL}}(D_\phi(\mathbf{a}, \mathbf{x}) \parallel D_\psi(\hat{\mathbf{x}}, \mathbf{x})). \tag{2}$$

Then, given the trained image discriminator $D_\psi$, we train the compression model using the standard GAN generator loss [22, 31],

$$\ell(\theta) = \sum_{\mathbf{x}\in\mathcal{D}} -\log D_\psi(f_\theta(\mathbf{x}), \mathbf{x}), \tag{3}$$

where $f_\theta(\mathbf{x})$ denotes $\mathbb{E}_{\hat{\mathbf{x}}\sim f_\theta(\hat{\mathbf{x}}|\mathbf{x})}[\hat{\mathbf{x}}|\mathbf{x}]$. Our complete pragmatic compression method is summarized in Algorithm 1. We randomly initialize the compression model $f_\theta$. The environment samples an original image $\mathbf{x}$ from an unknown distribution $p_{\text{env}}$. To decide whether the user sees the original or

compressed image, we sample a Bernoulli random variable $T$. After seeing the chosen image, the user samples an action $\mathbf{a}$ from their unknown policy $\pi$. To update the action discriminator $D_\phi$, we take a gradient step on the binary cross-entropy loss. To update the image discriminator $D_\psi$, we take a gradient step on the KL-divergence loss in Equation 2. To update the compression model $f_\theta$, we take a gradient step on the GAN generator loss in Equation 3. See Appendix A.3 for details.

## 4 Structured Compression using Generative Models

One approach to representing the compression model $f_\theta$ could be to structure it as a variational autoencoder (VAE) [28], and train the VAE end to end on the adversarial loss function in Equation 3 instead of the standard reconstruction error loss. This approach is fully general, but requires training a separate model for each desired bitrate (which is determined by the $\beta$ coefficient in the VAE training objective), and can require extensive exploration of the pixel output space before it discovers an effective compression model. To simplify variable-rate compression and exploration in our experiments, we forgo end-to-end training, and first train a generative model on a batch of images without the human in the loop by optimizing a task-agnostic perceptual loss, yielding an encoder and decoder such that $\mathbf{z} = \mathrm{enc}(\mathbf{x})$ and $\hat{\mathbf{x}} = \mathrm{dec}(\mathbf{z})$, where $\mathbf{z} \in \mathbb{R}^d$ is the latent embedding. Analogous to prior work on conditional image generation [32], we then train our compression model $f_\theta(\hat{\mathbf{z}}|\mathbf{z})$ to compress the latent embedding, instead of compressing the original pixels. We use a variety of different generative models in our experiments, including a $\beta$-VAE [33] for the handwritten digit identification experiments in Figure 3, a StyleGAN2 model [34] for the car shopping and survey experiments in Figure 4, an NVAE model [35] for the photo verification experiments in Figure 5, and a VAE for the car racing experiments in Figure 6. See Appendix A.4 for details.

Generative models like the VAE and StyleGAN2 tend to learn disentangled features – hence, instead of training $f_\theta$ to map directly to the latent space $\mathbb{R}^d$, we structure $f_\theta$ to output a vector of probabilities that determines which latent features are transmitted exactly between $\mathbf{z}$ and $\hat{\mathbf{z}}$, and which other features are masked out and then reconstructed from the prior distribution. In particular, we structure $f_\theta : \mathbb{R}^d \mapsto [0, 1]^d$ to output a vector of mask probabilities $\boldsymbol{p} \in [0, 1]^d$ given the latent embedding $\mathbf{z} \in \mathbb{R}^d$. Then, given a hyperparameter $\lambda \in [0, 1]$ that controls the compression rate, we transmit the $\lfloor \lambda d \rfloor$ latent features $i$ with the lowest mask probabilities $\boldsymbol{p}_i$, and mask out the remaining $d - \lfloor \lambda d \rfloor$ features. We reconstruct the masked features by assuming that $\hat{\mathbf{z}}$ follows a multivariate normal distribution, and sampling the masked feature values from the conditional prior distribution given the transmitted feature values. See Appendix A.4 for details.

This design of the compression model $f_\theta$ simplifies variable-rate compression: at test time, we simply choose a value of $\lambda$ that obtains the desired bitrate, without retraining the model. It also simplifies exploration: instead of exploring in pixel output space, we explore in the space of masks over latent features, which leverages the decoder to generate more realistic compressed images during the early stages of training. We can now also reduce the dimensionality of the image discriminator inputs: instead of training $D_\psi(\hat{\mathbf{x}}, \mathbf{x})$, we train $D_\psi(\boldsymbol{p}, \mathbf{x})$. In our experiments, we also leverage the low-dimensional mask output space to perform batch learning instead of online learning, which greatly simplifies our implementation of PICO with real users. See Appendix A.1 for additional discussion.

While these simplifications enable us to provide a proof of concept for pragmatic compression in this paper, we acknowledge that they do require both server and client to have a copy of a domain-specific (but task-agnostic) generative model. End-to-end training of the compression model would be a more general approach that does not involve learning and storing a separate generative model – this is a promising direction for future work, which we discuss in Section 6.

## 5 User Studies

In our experiments, we evaluate to what extent PICO can minimize the number of bits needed to transmit an image, while still preserving the image's usefulness to users performing downstream tasks. We conduct user studies on Amazon Mechanical Turk, in which we ask human participants to complete three tasks at varying bitrates: reading handwritten digits from the MNIST dataset [36], verifying attributes of faces from the CelebA dataset [37], and browsing a shopping catalogue of cars from the LSUN Car dataset [38]. To study PICO's performance on sequential decision-making

problems, we also run an experiment with 12 participants who play the Car Racing video game from OpenAI Gym [39] under a constraint on the bitrate of the video feed. In all experiments, we train our discriminators and compression model on 1000 negative examples and varying numbers of positive examples, and split PICO into two rounds of batch learning and evaluation (see Appendices A.1 and A.5). Appendix A discusses the implementation details.

## 5.1 Minimizing Bitrate by Maximizing User Action Agreement

We claim that PICO can learn to transmit only the features that users need to perform their tasks. Our first set of user studies seeks to answer **Q1**: does maximizing user action agreement enable PICO to obtain lower bitrates than baseline methods that do not take into account downstream user behavior? We would like to study this question in domains where we can measure the performance of various compression methods by computing the agreement between the user's actions with and without compression – i.e., collecting action labels for the original images, and comparing the user's actions upon seeing compressed versions of those images to the labels. As such, we run experiments on Amazon Mechanical Turk that focus on single-step decision-making settings where we can collect action labels for a fixed dataset of images: (a) identifying a handwritten digit, (b) clicking on an item in a shopping catalogue, and (c) verifying photos of faces. In (a), we instruct users to identify the number in the image within the range 0-9. In (b), to simulate the experience of browsing a catalogue on a budget, we instructed users to click on images of cars that they perceive to be worth less than $20,000. In (c), we instruct users to check if the person's eyes are covered (e.g., by eyeglasses) and click on one of two buttons labeled "covered" and "not covered".

In all domains, we evaluate PICO by varying the bitrate and, at each bitrate, measuring the agreement of user actions upon seeing a compressed image with user actions upon seeing the original version of that image (see Appendix A.7 for details). As discussed in Section 4, PICO learns a compression model $f_\theta$ that, given a separate generative model, selects which latent features to transmit. Since the purpose of this experiment is to test the effect of user-adaptive compression in PICO, we compare to a non-adaptive baseline method that selects a uniform-random subset of features to transmit, but otherwise uses the same generative model as PICO – this enables us to conduct an apples-to-apples comparison that isolates the effect of training $f_\theta$ on user behavior data. We also compare to a baseline method that maximizes perceptual similarity by replacing the adversarial loss in Equation 3 of PICO with the mean absolute pixel difference $|\mathbf{x} - \hat{\mathbf{x}}|$. In simulation experiments, we found that this perceptual similarity baseline performed better than the non-adaptive baseline in the MNIST domain, but did not perform better in the other domains (see Appendix C), so we only test it in the MNIST user study. To provide a point of comparison to widely-used compression methods, we also compare to JPEG [12], where the quality parameter is set to the lowest value (1) in order to bring the bitrate as close as possible to the range obtained by PICO and the non-adaptive baseline.

Though JPEG is no longer the state of the art, it enables us to roughly calibrate the results achieved by PICO as well as the non-adaptive and perceptual similarity baselines.

Figures 3, 4, and 5 show that, at low bitrates, PICO achieves substantially higher user action agreement than the non-adaptive baseline (orange vs. gray) and perceptual similarity baseline (orange vs. red). PICO also obtains much lower bitrates than the JPEG baseline (orange vs. teal), while maintaining higher agreement on CelebA, comparable agreement on MNIST, and lower agreement on LSUN Car. The samples in Figure 3 show that PICO learns to preserve digit numbers more often than the non-adaptive and perceptual similarity baselines, while randomizing handwriting

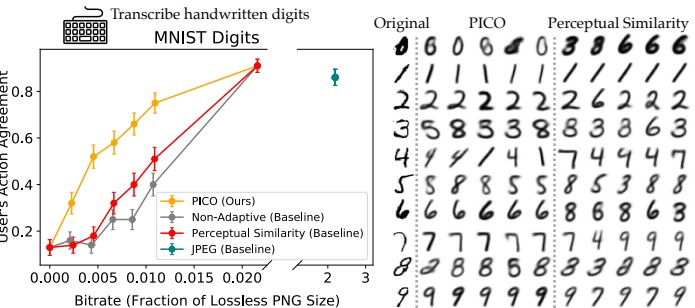

Figure 3: MNIST digit identification experiments that address **Q1**. When users are instructed to identify the digit number, PICO learns to preserve the digit number while randomizing handwriting style. The plots show user action agreement evaluated on 100 held-out images, with error bars representing standard error. The average lossless PNG file size is 0.3kB, and each image has dimensions 28x28x1. Each of the five columns in the two groups of compressed images represents a different sample from the stochastic compression model $f(\hat{\mathbf{x}}|\mathbf{x})$ at bitrate 0.011.

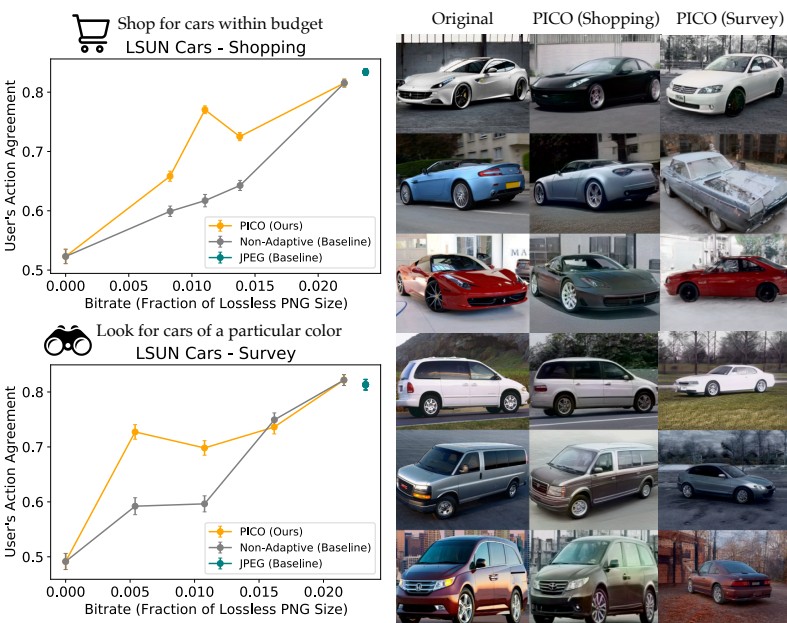

Figure 4: LSUN Car shopping experiment that addresses **Q1**, and survey experiment that addresses **Q2**. The plots show action agreement evaluated on 100 held-out images, with error bars representing standard error. The average lossless PNG file size is 247kB, and each image has dimensions 512x512x3. The shopping samples show that, when users are instructed to click on cars they perceive to be worth less than $20,000, PICO learns to preserve the overall shape and sportiness of the car, while randomizing paint jobs, backgrounds, and other details that are irrelevant to the users' perception of price. In contrast, when users are instead instructed to determine whether the car is "dark-colored" or "light-colored" for a survey task, PICO learns to preserve the car's color while randomizing its pose. We intentionally show compressed image samples for a low bitrate (0.011) to highlight differences between the compression models learned for the two tasks.

style in order to satisfy the bitrate constraint. The samples in Figure 4 show that, for users performing the shopping task, PICO learns to preserve the overall shape and sportiness of the car, while randomizing paint jobs, backgrounds, and other details that are irrelevant to the user's perception of the price of the car. The samples in Figure 5 show that, for users checking whether eyes are covered, PICO learns to preserve the presence of eyeglasses while randomizing hair color, faces, and other irrelevant details (see top row of samples). The dip in the orange curve in the car shopping plot may be due to the fact that increasing the bitrate preserves more of the encoded latent features, which, when combined with features sampled from the prior, can be out-of-distribution inputs to the StyleGAN2 decoder [40, 41], potentially leading to degraded image quality (see Appendix A.4 for details). Figures 9 and 10 in the appendix include more examples.

## 5.2 Adapting Compression to Different Downstream Tasks

The experiments in the previous section show that PICO can outperform a non-adaptive baseline method by transmitting only the features that users need to perform their tasks. Our second set of user studies investigates this mechanism further, by asking **Q2**: can PICO adapt the compression model to the specific needs of different downstream tasks in the same domain? To answer this question, we run an additional experiment in the CelebA domain from the previous section, in which users are instructed to check if the person's head is covered (e.g., by a hat). We also run an additional experiment in the LSUN Car domain from the previous section, in which we simulate a survey task that asks users to 'help a car dealership conduct market research' by determining whether an observed car has a "dark-colored" or "light-colored" paint job.

Figure 5 shows that PICO adapted the compression model to the user's particular task. In the experiment from the previous section, when users checked eyes, PICO learned to preserve the presence of eyeglasses while randomizing hair color, faces, and other irrelevant details (see top row of samples). On the other hand, when users checked for head coverings like hats and helmets, PICO

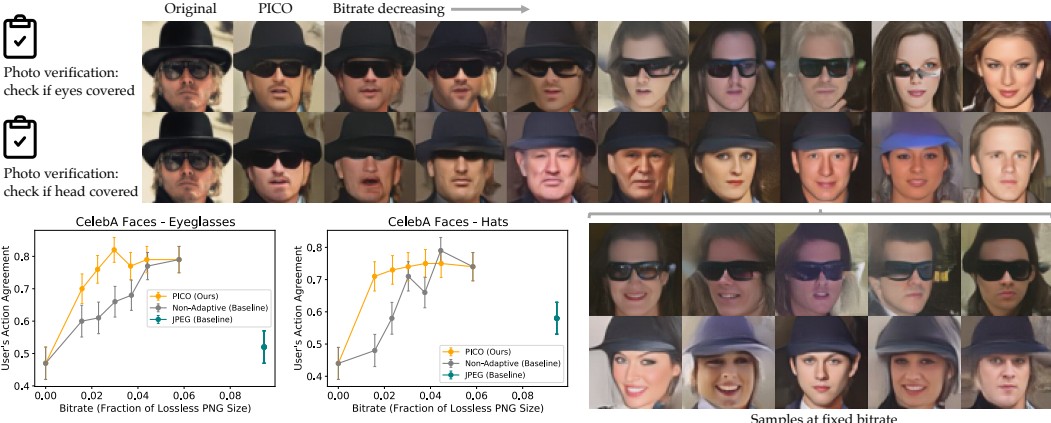

Figure 5: CelebA photo attribute verification experiments that address **Q1** and **Q2**. Depending on the instructions given to the user, PICO learns to either preserve hats or eyeglasses, while randomizing faces and other task-irrelevant details. The plots show action agreement evaluated on 100 held-out images, with error bars representing standard error. The average lossless PNG file size is 7.7kB, and each image has dimensions 64x64x3.

learned to preserve the presence of hats while randomizing eyes and other details (see second row of samples). The third and fourth rows of samples illustrate the fact that PICO learns a stochastic compression model $f_\theta(\hat{\mathbf{x}}|\mathbf{x})$ from which we can draw multiple compressed samples $\hat{\mathbf{x}}$ for a given original $\mathbf{x}$. The fact that all the samples in the third row have eyeglasses but differ in other attributes like pose angle, and those in the fourth row all have hats while some are smiling and some are not, shows that even though the compression model is stochastic, it produces stable attributes when they are needed for the downstream task. Figure 9 in the appendix includes more qualitative examples. In addition to these photo verification results, the samples in Figure 4 illustrate substantial differences in the compression models learned for the car shopping and survey tasks. For users performing the shopping task, PICO learned to preserve perceived price while randomizing color. In contrast, for users performing the survey, PICO learned to preserve color while randomizing perceived price.

### 5.3 Compressing Observations for Sequential Decision-Making

Our third user study seeks to answer **Q3**: can PICO learn to compress image observations in the sequential decision-making setting? To answer this question, we run an experiment with 12 participants in which we ask users to play a 2D top-down car racing video game, while constraining the number of bits that can be used to transmit the image observation to the user at each timestep. We would like to measure the performance of PICO and the non-adaptive baseline by computing user action agreement, as in the previous sections. However, since images rarely re-occur in this video game, it is unlikely that we will have an action label for the exact pixels in any given observation. Instead, we measure the user's progress along the road in the game – specifically, the fraction of new road patches visited during an episode. In these experiments, we fix the bitrate to 85 bits per step, which is well below the 170 bits per step required to transmit the full set of features for the 64x64x3 images. To simplify our experiments and ensure that they could be completed within the allotted 30 minutes per participant, we trained the PICO compression model on data from a pilot user, then evaluated the compression model's performance with each of the 12 participants. Appendix A.7 describes the experimental setup in further detail.

Figure 6 shows that, at a fixed bitrate, PICO enables the user to perform substantially better on the driving task than the non-adaptive compression baseline (orange vs. gray), and comparably to a positive control in which we do not compress the image observations at all (orange vs. teal). The first and second film strips show that, when we use the non-adaptive compression baseline, there is a substantial difference between the originals and the compressed images. For example, even at the first timestep, the compressed image shows the road to be less tilted than it actually is, so in the next frame we see that the user has mistakenly driven forward and ended up in the grass instead of turning right to stay on the road. In contrast, the third and fourth film strips show that PICO has learned to preserve the angle of the road, while discarding the details of the road much farther ahead in order to satisfy the bitrate constraint. We ran a one-way repeated measures ANOVA on the road progress

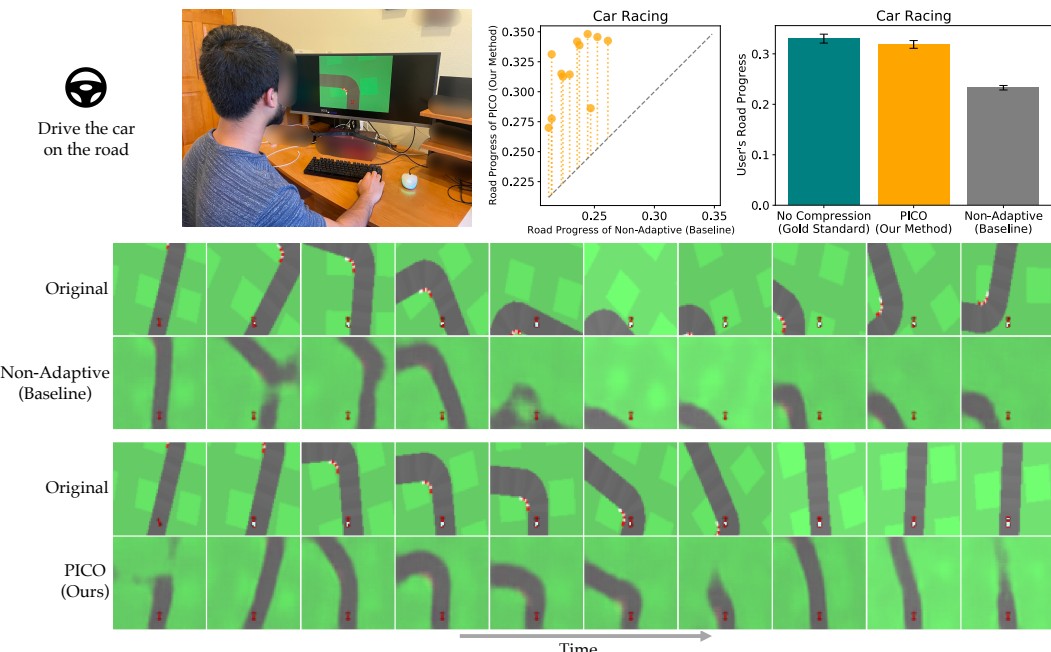

Figure 6: Car Racing game experiments that address **Q3**. The scatter plot shows that, for each of the 12 users (orange), road progress with PICO was substantially higher than with the non-adaptive compression baseline. The bar chart shows road progress averaged over all users, with error bars representing standard error.

metrics from the non-adaptive baseline and PICO conditions with the presence of PICO as a factor, and found that $f(1, 11) = 176.32, p < .0001$. The subjective evaluations in Table 1 in the appendix corroborate these results: users reported feeling higher situational awareness and ability to control the car with PICO compared to the non-adaptive baseline. After evaluating PICO, one user commented, "This environment was a lot easier. It felt more consistent. I felt like we had a mutual understanding of when I would turn and what it would show me to make me turn." Appendix B discusses the results in more detail, and videos are available on the project website[1].

# 6  Discussion

We presented a proof of concept that, through human-in-the-loop learning, we can train models to communicate relevant information to users under network bandwidth constraints, without prior knowledge of the users' desired tasks. Our experiments show that, for a variety of tasks with different kinds of images, pragmatic compression can reduce bitrates 2-4x compared to non-adaptive and perceptual similarity baseline methods, by optimizing reconstructions for functional similarity. Since we needed to carry out user studies with real human participants, we decided to limit the number of parameters trained during these experiments for the sake of efficiency, by using a pre-trained generative model as a starting point and only optimizing over the latent space of this model. This can be problematic when the generative model does not include task-relevant features in its latent space – e.g., the yellow sports car in rows 7-8 of Figure 10 in the appendix gets distorted when encoded into the StyleGAN2 latent space, even without any additional compression. An end-to-end version of PICO should in principle also be possible, but would likely require longer human-in-the-loop training sessions. This may, however, be practical for real-world web services and other applications, where users already continually interact with the system and A/B testing is standard practice. End-to-end training could also enable PICO to be applied to problems other than compression, such as image captioning for visually-impaired users, or audio visualization for hearing-impaired users [42] – such applications could also be enabled through continued improvements to generative models for video [43, 44], audio [45], and text [46, 47]. Another exciting area for future work is to apply pragmatic compression to a wider range of realistic applications, including video compression for robotic space

---

[1] https://sites.google.com/view/pragmatic-compression

exploration [13], audio compression for hearing aids [48, 49], and spatial compression for virtual reality [50].

## 7 Acknowledgements

Thanks to members of the InterACT and RAIL labs at UC Berkeley for feedback on this project. This work was supported in part by AFOSR FA9550-17-1-0308, NSF NRI 1734633, GPU donations from NVIDIA, and the Berkeley Existential Risk Initiative.

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
