# OpenReview forum: "Pragmatic Image Compression for Human-in-the-Loop Decision-Making"
_NeurIPS.cc/2021/Conference — NeurIPS 2021 Spotlight_

### Official Review · Reviewer_cV9i · 2021-07-16

**Rating:** 7
**Confidence:** 4

**Summary:**

This paper propose a deep image compression method that train a compression model through human-in-the-loop.
The method train discriminator that tries to distinguish whether a user’s action was taken in response to the compressed image or the original. They evaluate their method through tasks that do not rely solely on perceptual similarity.

**Limitations And Societal Impact:**

I thought it would be good to discuss whether having a person in the learning loop could potentially introduce bias into the compression results.

**Main Review:**

I think this paper is interesting challenge to the limit of bit rate by taking advantage of the characteristics of deep image compression.
Using action discriminator and distilled image discriminator for training the image compression model may be a new approach.
Also, I think it would be useful for readers to show how much learning time is required for human-in-the-loop learning compared to baseline.
One minor concern is that the bitrate is expressed in comparison to PNG and not in absolute bitrate (e.g. bit-per-pixel), which makes it difficult to get a sense of the bitrate level.

**Time Spent Reviewing:**

10

---

> ### Author Response · Authors · 2021-08-06
> **Response**
>
> Thank you for the thoughtful and constructive feedback. We will include learning times and absolute bitrates in the final version of the paper. We will also add a discussion of the potential biases introduced by human-in-the-loop learning.

---

> > ### Comment · Reviewer_cV9i · 2021-08-26
> > **Thank you for your response.**
> >
> > I think the approach of this paper is interesting and novel.
> > Your responses have addressed some of my concerns and I will raising my rating.
> > I think that the paper should be a start point to explore the good use cases of this new compression approach.

---

### Official Review · Reviewer_SBfJ · 2021-07-16

**Rating:** 6
**Confidence:** 4

**Summary:**

The paper proposes a downstream task specific compression method by training a compression model through human-in-the-loop leaning, which adaptively maintain core information based on whether human can make the same decisions when they see compressed or original images. The user study show the proposed method outperforms JPEG codec.

**Limitations And Societal Impact:**

Discussed but not adequately addressed. Some constructive suggestions can be found above.

**Main Review:**

Originality: Novel but not good. The human-in-the-loop learning is firstly used in compression to the best of my knowledge. The proposed method directly evaluates the quality of images based on the consistency of human decisions seeing compressed/original images. And the distillation design, using another discriminator which judge images with action to train the wanted discriminator which judge images directly, is brilliant.
However, this design hugely relies on participants, which limits its generalization and evaluation. In fact, it replaces a lot of concurrent perceptual loss functions with “human-in-the-loop” in my opinion.

Quality: Bad. The drawback of this work is intuitive, that is, it requires a lot of users to infer which actions to take, and it is time-consuming because with the model trained, pictures provided to users change as well. The paper discusses this in Sec. 6 but I do not totally agree that using web services is a good solution, which means the model will be hugely affected by users during a period, which can be biased in my opinion.
On the other hand, whether the “human” in the loop can be replaced with a neural network, which just requires ground truth label and it can be collected at a time, is not discussed. The excellent performance of neural network to simulate certain functions should be considered, and I think adding some module like attention map will improve the performance.
Furthermore, there is a lot of advanced compression methods besides relatively old JPEG. More experiments for comparison should be provided. And user study data of JPEG is not enough at all. I want to see its performance with low bit-rate.
To summarize, I doubt that the proposed hugely participant-oriented method is a good solution to so-called Pragmatic Image Compression. And if we just consider the performance of some specific task, why not directly training a network to tell users the label they wanted, without image at all (both ways need different models for different tasks and label needs slight bit-rate).

Clarity: Could be better. The human-in-the-loop strategy can be explained earlier and more clearly, which is a key problem to discuss. The References should be arranged in a uniform style.

Significance: Not good. The paper does not provide enough data of JPEG with low bit-rate. As we can notice in Fig. 3, number 0, 3, 5 are not reconstructed enough and I doubt that some autoencoder-based methods can surpass it.


**Time Spent Reviewing:**

3 hours

---

> ### Author Response · Authors · 2021-08-06
> **Response**
>
> Thank you for the thoughtful and constructive feedback.
>
> > On the other hand, whether the “human” in the loop can be replaced with a neural network, which just requires ground truth label and it can be collected at a time, is not discussed. The excellent performance of neural network to simulate certain functions should be considered, and I think adding some module like attention map will improve the performance.
>
> Figure 7 in Appendix C includes experiments where we train a neural network to simulate user behavior given ground truth action labels, instead of training with a real human in the loop. The results show that our method outperforms the baselines even when trained with a user model instead of real users. We apologize for the miscommunication, and will add a pointer to this appendix in the main paper.
>
> > However, this design hugely relies on participants, which limits its generalization and evaluation.
> > The drawback of this work is intuitive, that is, it requires a lot of users to infer which actions to take, and it is time-consuming because with the model trained, pictures provided to users change as well.
>
> Our method learns from human-in-the-loop data that is naturally generated as users perform their day-to-day tasks (e.g., online shopping, or remotely operating robots), and does not require the users to alter their natural behavior or pursue artificial goals specified by our system. Thus, our method fits into existing A/B testing frameworks that are already used by many popular web services to optimize recommendation systems, search result ranking algorithms, and other automated systems. These A/B testing frameworks have already been designed to minimize interference with users’ experiences, and can be used to implement our method.
>
> Furthermore, when used with a generative model, our human-in-the-loop learning method is relatively sample-efficient. As mentioned in the 1st paragraph of Section 5, our method learns from only 1000 negative examples of user behavior. These examples took less than one hour to collect on Amazon Mechanical Turk. Our experiments suggest that our method could potentially be scaled up to larger applications without requiring impractical amounts of human-in-the-loop training.
>
> > And user study data of JPEG is not enough at all. I want to see its performance with low bit-rate.
> > The paper does not provide enough data of JPEG with low bit-rate.
>
> As mentioned in the 2nd paragraph of Section 5.1, we set the quality parameter of JPEG to its lowest possible value of 1 in order to achieve the lowest possible bitrate. This lowest-possible bitrate was still larger than the highest bitrate possible with our method (which is achieved by transmitting all the latent features of the original image). We will add examples of images compressed by JPEG with quality=1 to the final version of the paper.

---

> > ### Author Response · Authors · 2021-08-14
> > **Request for feedback**
> >
> > Thank you again for the thoughtful review. We would like to know if our rebuttal adequately addressed your concerns. Are there any other aspects of the paper that you think could be improved?

---

> > > ### Comment · Reviewer_SBfJ · 2021-08-20
> > > **Score can be lifted if my concern is well solved**
> > >
> > > Thanks a lot to your detailed clarification and explanations.
> > >
> > > After reading your Response and other reviews, I would like to rethink my review and final score. In my opinion, this work provide a novel and elegant solution to human-centered tasks, which is a consensus among reviewers.
> > >
> > > My main concern falls on its real application. From another perspective, does the image decoded from compact representation really maintain what users focus?
> > >
> > > If we focus on Decision-Making only, I want to see an experiment *i.e.* encoder network analyses the original image and do the Decision-Making directly with the same adversarial way, and then the predicted decision will be compressed and send to the user, which means users just directly see a decision recommended.
> > >
> > > I want to discuss whether the aforementioned way has vital difference if we just focus on Decision-Making and regard all other information useless. And if there is a good solution, I would like to lift my score very much.

---

> > > > ### Author Response · Authors · 2021-08-20
> > > > **New experimental results**
> > > >
> > > > Thank you for your response! Today, we ran the experiment you suggested: we modified the MNIST experiment with simulated users in Appendix C such that, instead of showing the user a compressed image, we simply show the user our compression model's predicted class probabilities (i.e., a 10-dimensional vector). We kept all the other details of our algorithm the same. We found that, with a bitrate of only $\log_2{10} \approx 3.3$ bits, our method enables the simulated user to achieve 96% action agreement, where the random baseline method only achieves 10% action agreement. This result shows that, in a single-task setting, when we structure the output space of our compression model to represent decisions instead of images, our method can learn to recommend decisions directly to the user.

---

> > > > > ### Author Response · Authors · 2021-08-26
> > > > > **Request for feedback**
> > > > >
> > > > > We would appreciate it if you could let us know if there are any other experiments we can run that would change your score. Only one more week remains of the discussion period, and we would like to ensure that you and the AC have all the information you need from us. Thank you!

---

> > > > > > ### Comment · Reviewer_SBfJ · 2021-08-31
> > > > > > **Feedback**
> > > > > >
> > > > > > I'm sorry to be so late to respond to you. Thank you for your new experiments and they partly addressed my concerns.
> > > > > > I would like to change my original score to ACCEPT the paper. Meanwhile, I still think it is a critical problem that, if we just focus on whether users can make "correct" decisions, do we need to show them images? Instead, we can show them some recommendations, which are very compact and the final agreement performance is also great.
> > > > > > It is a hard question to answer and I do not require author(s) answers it immediately. The work can be a good start. :)
> > > > > > Thank you.

---

> > > > > > > ### Author Response · Authors · 2021-08-31
> > > > > > > **Follow up**
> > > > > > >
> > > > > > > Thank you for your response! Could you please edit the numerical score in your original review to reflect your updated recommendation to accept the paper?

---

### Official Review · Reviewer_99WX · 2021-07-16

**Rating:** 8
**Confidence:** 5

**Summary:**

The authors claim to account not just for optimizing a lossy image compression algorithm to preserve an image's appearance but additionally to minimizes bits while maintaining information necessary for a user to complete downstream tasks correctly.  They achieve this by training a discriminator with human in the loop training and demonstrate the effectiveness of their methodology.  They claim that they show that the method learns to match the user's actions with and without compression at lower bitrates than baseline methods and adapts the compression to user behavior.

**Limitations And Societal Impact:**

Could be used to illicit what parts of images or other inputs human users are utilizing to make their decisions without the user realizing that is happening, not just in toy tasks but in other everyday interactions with things like phones and other devices. Though the method would have to progress quite significantly before that would become a real worry. The authors did not discuss this though they may not have realized the impact for implicit information extraction that is available via a method like this.

**Main Review:**

This is a clever paper that uses a novel combination of methodologies to achieve its results. These authors are not the first to train a compression algorithm for downstream tasks as an objective, but as far as I can tell, they are the first to have those downstream tasks completed by real users in the training loop, such that the compression algorithm is learning to optimize for real users and not some modeled approximation to them, or to some CV algorithm.   They call this PICO (pragmatic compression).

The results themselves (though meeting the authors claims) don't suggest to me that they are extremely useful as an application as of yet (ie you wouldn't surface these images via a web tool for anyone to actually complete the task if you were relying on that task being done perfectly).  However, the key application (not fleshed out by the authors) seems to me to be in the cognitive sciences, as this serves as a powerful engine for pulling out of images the key elements that people are relying on to do the downstream tasks.

Overall I think this is a novel and cleve contribution that will be interesting to many people at NeurIPS.


**Time Spent Reviewing:**

2

---

> ### Author Response · Authors · 2021-08-06
> **Response**
>
> Thank you for the thoughtful and constructive feedback. We are excited to look into potential applications of PICO in cognitive science experiments that seek to understand the visual features that humans rely on to make decisions.

---

### Official Review · Reviewer_Maz5 · 2021-07-17

**Rating:** 8
**Confidence:** 3

**Summary:**

The foundational motivation for this work is the quite reasonable assumption that when a learned image compression codec is to be used as a component in a larger system, it might be desirable to optimize the codec for its functional utility for the downstream task instead of general perceptual quality. Forgoing the need for good all-around perceptual reconstruction can therefore lead to large gains in terms of compression ratio.

The main challenge is to quantify the functional similarity of a compressed image compared to the original image, without making any assumptions about what the users' downstream task is, or how the users select their actions upon seeing an image. To this end, the authors propose an adversarial framework. In a manner similar to A/B testing, they present users with either the original images $x$ or their compressed versions $\hat{x}$, and record what actions $a$ the users took. Then, they train a discriminator $D(a, x)$ to predict if a user's action was taken after seeing the original or the compressed image. This first discriminator is distilled to a second discriminator $D(x, \hat{x})$, which predicts which image was seen based on the prediction of the first discriminator. Now, the second discriminator $D(x, \hat{x})$ can be readily used as a measure of functional similarity between $x$ and $\hat{x}$ for the unknown downstream task of the user.

If this system was to be trained end-to-end, it would require access to a large amount of data harvested from the users. To circumvent this, the authors use pre-trained generative models in their experiments, and optimize the discriminators on quantities derived from the generative models' latent spaces, making the optimization process converge much faster.

The authors perform extensive experiments to verify that 1) their proposed adaptive method performs better than general-purpose, non-adaptive methods, 2) their method adapts to the specific needs of the downstream task and 3) their method can learn in a sequential decision-making setting. They obtain good results for all 3 settings.


**Limitations And Societal Impact:**

They have.

**Main Review:**

# Strengths
The work is well-motivated, the problem and the framework are interesting, and the utility of a solution is clear. The use of the adversarial framework to obtain a proxy metric for the functional similarity between the original and compressed images is a simple and elegant solution.

The reduction of the problem to the latent space of generative models is also elegant and greatly extends the applicability of the proposed method.

The experiments are well-motivated, well-set-up and well-executed. The illustration and analysis of which features PICO retains or allows to vary is very convincing.

The paper is well-written with good structure. Very nice submission overall.

# Weaknesses
I am slightly confused about the "bitrate-user action agreement" graphs. Concretely, I don't understand how the methods can start at 0 bitrate. If the bitrate is actually a positive number close to 0, then this should be cleared up as the graphs are confusing/ misleading in their current state.

# Questions
- Have the authors attempted end-to-end training their method on toy tasks instead of reducing it to the latent space of a generative model? It seems to me that one reason the proposed method works so well is because the general-purpose compression codec provided by the generative model is already quite good.
- In particular, one reason for such a clear behaviour of the system to keep relevant features and randomize irrelevant ones is due to the fact that the authors used pre-trained generative models that are known to be biassed towards learning independent latent features. Do the authors think we would see a similar disentanglement if they trained the system end-to-end?


**Time Spent Reviewing:**

4-6

---

> ### Author Response · Authors · 2021-08-06
> **Response**
>
> Thank you for the thoughtful and constructive feedback.
>
> Weaknesses:
>  - In our experiments, a bitrate of zero corresponds to not sending any information to the client. In this case, the client randomly samples an image from the prior distribution of the generative model, and shows this random image to the user. Of course, this condition is not particularly interesting, and it is included in the graphs mainly as a control. We will clarify and explain this in the final version.
>
> Questions:
>  - We have not yet fully tested end-to-end training on toy tasks. We will work towards this for the final version of the paper.
>  - Although we have not tested this, it should be possible to combine the end-to-end version of our method with complementary methods for inducing disentangled latent features, such as the information bottleneck. That said, we agree that using a pretrained model likely makes the learning problem much easier, and we will add a discussion of this point to the paper.

---

> > ### Comment · Reviewer_Maz5 · 2021-08-31
> > **Response**
> >
> > I thank the authors for their response and their answers to my questions. I have been impressed with the work in general as well as the authors' rebuttal to the reviews, hence I keep my recommendation for acceptance.

---

### Decision · Program_Chairs · 2021-09-27

**Decision:**

Accept (Spotlight)

**Comment:**

Strengths:
- Novel and elegant approach
- Clever training procedure that incorporates humans in-the-loop
- Can potentially give insight into human decision-making process (from visual stimuli)
- Well-executed experiments involving real human subject feedback


Weaknesses:
- No clear application or real-world use case
- Requires a more careful discussion on algorithmic decision support and agency


Summary:

Reviewers were mostly unanimous in their opinion that this is a solid contribution to the growing literature on learning to support human decision-making. One reviewer initially raised several concerns regarding practicality, but the authors response (which included additional experimentation) were helpful in increasing his score.

Learning with humans in-the-loop is challenging, and this paper does a commendable job at executing experiments in this setting. As one reviewer comments, the authors should clarify the amount of human resources (time, queries, etc.) required in their approach. But more importantly, I would encourage the authors to discuss in detail the potential impacts of deploying their algorithm in the real world. Issues such as the role of algorithms for decision-support and their relation to human agency should be acknowledged – especially given that the paper does not highlight any specific use-case. The authors should also carefully consider and justify what they measure and why. All in all, paper presents a fresh approach and is likely to be the focus of interesting follow-up work.